



# 1  Eurasian snow depth in long-term climate
# 2  reanalyses

Martin **Wegmann**[1,2,3], Yvan **Orsolini**[4], Emanuel **Dutra**[5], Olga **Bulygina**[6],
Alexander **Sterin**[6] and Stefan **Brönnimann**[2,3]
[1] *Institut des Géosciences de l´Environnement, University of Grenoble, France*
[2] *Oeschger Centre for Climate Change Research, University of Bern, Switzerland*
[3] *Institute of Geography, University of Bern, Switzerland*
[4] *NILU—Norwegian Institute for Air Research, Kjeller, Norway*
[5] *ECMWF European Centre for Medium-Range Weather Forecasts, Reading, UK*
[6] *All-Russian Research Institute of Hydrometeorological Information—World Data*
*Centre, Obninsk, Russian Federation*
*Corresponding Author:*
*Martin Wegmann, martin.wegmann@univ-grenoble-alpes.fr*





**Abstract**
Snow cover variability has significant effects on local and global climate evolution.
By changing surface energy fluxes and hydrological conditions, changes in snow
cover can alter atmospheric circulation and lead to remote climate effects. To analyze
such multi-scale climate effects, atmospheric reanalysis and derived products offer the
opportunity to analyze snow variability in great detail far back in time. So far only
little is know about their quality. Comparing four long-term reanalysis datasets with
Russian in situ snow depth data, a good representation of daily to sub-decadal snow
variability was found. However, the representation of pre-1950 inter-decadal snow
variability is questionable, since datasets divert towards different base states. Limited
availability of independent long-term snow data hinders investigating this bifurcation
of snow states in great detail, but initial investigations reveal a non-stationary
performance of snow evolution representation. This study demonstrates the ability of
long-term reanalysis to reproduce snow variability accordingly.















## 1. Introduction


Snow is an important component of the climate system over the mid- and high-
latitude regions of the Earth. Its high shortwave albedo and low heat conductivity
alters heat and radiation fluxes at the Earth´s surface and thus directly modulates
regional temperature evolution and ultimately atmospheric circulation patterns
(Barnett et al. 1988, Cohen and Rind 1991, Callaghan et al. 2011, Cohen et al. 2014).
Moreover, because snow acts as a temporary water reservoir, snow variability impacts
soil moisture, evaporation and ultimately precipitation processes (Yasunari et al.
56    1991).

As a result, snow cover has an essential influence on ecological (Jonas et al. 2008,
Peñuelas et al. 2009) and economical systems (eg. Agrawala 2007). Vice versa, snow
cover itself is determined by climate variations. Recent Arctic warming severely
impacted spring snow cover. Between 1979 to 2011, Arctic April snow cover extent
decreased at a rate of -17.8% per decade (Derksen and Brown 2012). In contrast,
regional snow cover increase in autumn over Eurasia was found in connection with
low Arctic sea ice concentration (Honda et al. 2009, Park et al. 2013, Wegmann et al.
2015), indicating the complexity of global and regional processes leading to snow
cover changes.
Reciprocally, as a slowly varying component of the climate system, the snow cover
influences large-scale climate patterns, and has been tapped as a source of
predictability at the subseasonal-to-seasonal scale, especially over Eurasia in autumn
and winter (Cohen and Entekhabi 1999, Jeong et al. 2013, Orsolini et al. 2013, Wu et
al. 2014, Ye et al. 2015,).
Therefore, large-scale monitoring and quantifying of snow cover is crucial for
assessing climate change and its representation in climate models (eg. Frei and Gong
2005, Brown and Mote 2009, Brown and Robinson 2011, Liston and Hiemstra 2011,
Ghatak et al. 2012, Zuo et al. 2015) and for analyzing cryosphere-climate feedbacks
(eg. Flanner et al. 2011, Orsolini and Kvamstø 2009, Zhang et al. 2013). Here we
analyze snow depths in climate in comparison to in-situ data, with the aim to better
assess cryosphere-atmosphere coupling processes in the context of the 20th century
climate evolution.



To this end, reanalysis products provide a compromise between the high temporal
resolution and length of in-situ observational datasets (eg. Bulygina et al. 2010) and
the large spatial coverage of satellite products (Siljamo and Hyvärinen 2011, Frei et al.
2012, Hüsler et al. 2014). Comprehensive reanalyses datasets are well suited to
investigate processes and mechanisms, and a variety of reanalyses are now routinely
produced by meteorological prediction centers, covering not only the satellite era but
also extending further back in time, such as (but not limited to) NCEP-DOE, ERA-40
and ERA-Interim, and JRA-25 and JRA-55 (e.g. Uppala et al. 2005, Onogi et al. 2007,
Compo et al. 2011, Dee et al. 2011, Rienecker et al. 2011, Poli et al. 2013).
However, so far only a few studies analyzed snow representation in reanalysis
products. Khan et al. (2008) compared measured snow data with snow water
equivalents and snow depth in the NCEP-DOE (Kanamitsu et al. 2002), ERA-40
(Uppala et al. 2005) and JRA-25 (Onogi et al 2007) reanalysis products over Russian
river basins. They found that the ERA-40 outperformed the NCEP-DOE and JRA25
in terms of correlations and mean values. Despite reproducing well the seasonal
variability, all reanalysis products struggled with snowmelt season values. Brown et al.
2010 compared ERA-40 and NCEP/NCAR snow cover extent to satellite and in-situ
datasets. They found that for the period 1982-2002 ERA-40 shows higher correlations
and smaller root mean squared errors (RMSE) than the NCEP reanalysis, and that
May values were considerably better approximated than June values. Brun et al. 2013
forced the CROCUS snow model with atmospheric conditions from ERA-INTERIM
(1970-1993) and found very high agreements with Eurasian in-situ snow
measurements. However, no snow output from the reanalysis directly was evaluated.
In addition, climate reanalyses extending back to the beginning of the 20th century or
earlier have now been produced for multi-decadal climate studies. Contrarily to the
above-mentioned reanalyses, these climate reanalyses, namely the 20th Century
Reanalysis (20CRv2) (Compo et al. 2011) and ERA-20C (Poli et al. 2016), solely rely
on assimilation of surface data. Even fewer studies have tried to quantify snow cover
extent and depth and their potential impact on climate in such centennial reanalyses.
Recently, Peings et al. 2013 compared in-situ snow measurements over Russia with
20CRv2 for the whole 20th century, and found that it consistently and realistically
represents the onset of Eurasian snow cover. However, the authors only investigated



the snow dataset in a binary fashion (snow/no snow).
Given the lack of inter-comparison studies of snow depth between reanalyses
products, we evaluate snow depth in four centennial state-of-the-art reanalyses. The
goal of this study is to assess the consistency between in-situ observations and
reanalyses estimation of snow depths. To assess this performance, we focus on early
snowfall season (October, November) and early snow melt season (April). Land
reanalyses will also be used in the assessment.
This article is structured as follows. Section 2 gives an overview of the various
datasets analyzed, whereas Section 3 defines the methods used in the comparison.
Section 4 presents the results for the evaluation. After discussing the results in Section
5, conclusions are drawn in Section 6.
**2. Data**
In this study, we use six different climate reanalysis datasets, which can be divided
into two families, namely the European Centre for medium-range Weather Forecasts
(ECMWF) products and the NOAA-CIRES Twentieth Century Reanalysis products.
These datasets are compared with Russian in-situ snow depth measurements.
**2.1 Reanalysis Datasets**
The Twentieth Century Reanalysis Version 2 (20CRv2) dataset allows retrospective
4-dimensional analysis of climate and weather between 1871 and 2012 (Compo et al.
2011). It was achieved by assimilating surface observations of synoptic pressure into
the NCEP GFS model using an Ensemble Kalman Filter variant. Prescribed boundary
conditions are HadISST1.1 (Rayner et al. 2003) monthly sea-surface temperature
(SST) and sea ice cover data as well as forcing of $CO_2$, volcanic aerosols and solar
radiation.
The 20th Century Reanalysis Version 2c (20CRv2c) uses the same model as version 2
with new sea ice boundary conditions from the COBE-SST2 (Hirahara et al. 2014),
new pentad Simple Ocean Data Assimilation with sparse input (SODAsi.2, Giese et al.
2015) sea surface temperature fields, and additional observations from ISPD version
3.2.9 (Cram et al. 2015). SODAsi2c is generated by tapering SODAsi.2 at 60° N/S to
COBE-SST2 SSTs, which makes the Arctic sea ice and SSTs consistent. For both



products, we use the mean of the 56-member ensemble, at a 6-hourly temporal
resolution. The spatial resolution corresponds to a Gaussian T62 grid.
The ERA-20C (ERA20C) reanalysis (Poli et al. 2016) uses the Integrated Forecast
System (IFS) model as a framework to assimilate observations of surface pressure and
marine surface winds. It is a global atmospheric reanalysis for the period 1900 – 2010
with a 3-hourly temporal resolution and a horizontal resolution of T159 with 91
vertical levels, reaching from the surface up to 1 Pa. Sea – ice cover and SST forcing
come from an ensemble of realizations (HadISST.2.0.0.0), where the variability in
these realizations is based on the uncertainties in the observational sources used for
this forcing. The radiation scheme follows exactly the CMIP5 proposal, including
aerosols, ozone and greenhouse gases (Hersbach et al. 2015).
In addition to the ERA20C reanalysis, the ERA-20C and ERA-Interim (1979-2015)
(Dee et al. 2011) land versions (Balsamo et al. 2015) (ERA20CL & ERA-INTERIM-
land) are used in our assessment. These land reanalyses consist of off-line runs of the
ECMWF land surface model, driven by the atmospheric forcing from the respective
reanalysis. When calculating the correlation and root-mean-square error, both the
corrected (with GPCP) and uncorrected version of ERA-INTERIM-land are used
(referred to ERAINTL-d and ERAINTL-e, respectively). For spatial plots, we only
show the corrected version. ERA20C was analyzed in 0.5° resolution, and ERA-
INTERIM-land in 1° resolution. It is important to note that none of the atmospheric or
land reanalyses used in this study assimilated snow measurements.

### 2.2 Snow depth observations

This study uses time series of daily snow depths for 820 Russian meteorological
stations (distributed as shown in the supplementary Figure 1). The time series are
prepared by RIHMI-WDC (All-Russian Research Institute of Hydrometeorological
Information—World Data Centre). Meteorological data sets are automatically
checked for quality control. Since the procedure of snow observations changed in the
past, particular attention was given to the removal of all possible sources of
inhomogeneity in the data. However, there have been no changes in the observation
procedures since 1965. When using monthly data, we use the maximum snow depth
during that month instead of mean value, because it reflects the process of snow



accumulation (snow depth is a cumulative and highly inertial characteristic of climate
system). It is especially essential for autumn months when the main processes of snow
accumulation occurs over the territories of Russia.
**3. Analysis procedure**
**3.1 Choice of long-term daily snow observations**
Out of the over 800 stations, 15 stations were selected with a record extending back to
the beginning of the 20th century on a daily basis. Stations with records extending
into the 19th century were shortened to start from 1901. All time series end in 2011.
Stations with different starting years are indicated in Table 1. Furthermore, Table 1
displays the location of the 15 stations, including the elevation above sea level. To
correlate daily measurements with daily reanalysis values, values from the closest grid
cell to the station location were chosen. Moreover, the relative amount of missing data
is shown for the each of the three months considered in this study. As can be seen,
data availability differs considerably between months and stations. However, one
station (ID 35108) exceeds 20% missing data in all three months was excluded from
further analysis. We also excluded one station (ID 32098) for which the related grid
box was classified as ocean. This results in a final selection of 13 stations.
**Table 1:** 15 long-term snow stations taken out of the Russian snow station data pool.
Listed are WMO ID, name, coordinates, elevation as well as starting year and missing
values. Missing values are indicated relative to the whole sample size of each
individual station for April (A), October (O) and November (N). Red marked stations
where excluded from further analysis.

| WMO ID | Station Name | Coordinates | Elevation above sea level | Starting year if not 1901 | Missing values in % |
|---|---|---|---|---|---|
| **22550** | Arhangel`sk | 64°30` N 40°44` E | 8 | | A (8.8) O (7.9) N (12) |
| **23405** | Ust`-Cil`ma | 65°26` N 52°16` E | 78 | 1914 | A (6.9) O (6.6) N (5.3) |
| **23711** | Troicko-Pecherskoe | 62°42` N 56°12` E | 135 | | A (5.5) O (6.6) N (6.3) |
| **24641** | Viljujsk | 63°47` N 121°37` E | 110 | 1903 | A (13.5) O (21) N (17.4) |
| **24966** | Ust`-Maja | 60°23` N 134°27` E | 169 | | A (16.1) O (17) N (17.2) |



| | | | | | |
|---|---|---|---|---|---|
| **26063** | St. Petersburg | 59°23` N 30°18` E | 3 | 1902 | A (9.2) O (8) N (16.6) |
| **27199** | Kirov | 58°36` N 49°38` E | 157 | | A (10.4) O (10.6) N (14) |
| **27675** | Poreckoe | 55°11` N 46°20` E | 136 | | A (17.5) O (11.7) N (23.2) |
| **27955** | Samara (Bezencuk) | 52°59` N 49°26` E | 45 | 1904 | A (7.7) O (3.5) N (11.3) |
| **28275** | Tobol`sk | 58°09` N 68°15` E | 49 | 1907 | A (17.1) O (17.4) N (23.2) |
| **28440** | Ekaterinburg | 56°50` N 60°38` E | 281 | | A (5.6) O (2.5) N (3.3) |
| **30758** | Chita | 52°05` N 113°29` E | 671 | 1926 | A (8.3) O (8.1) N (10.4) |
| **32098** | Poronajsk | 49°13` N 143°06` E | 7 | 1908 | A (3.2) O (2) N (8.4) |
| **35108** | Urals (Kazakhstan) | 51°15` N 51°17` E | 37 | | A (21) O (24.7) N (30.8) |
| **35121** | Orenburg | 51°41` N 55°06` E | 115 | | A (5.4) O (7.9) N (13.1) |



## 3.2 Calculation of extreme event detection

To evaluate the detection rate of extreme daily snow depth events, we calculate the 98th percentile values in all reanalysis products in two different ways. Extreme events were calculated for both absolute snow depth and accumulated snow depth, the later being the snow depth difference between two consecutive days. The selected dates in the reanalyses are then compared to the station dates. Based on the number of dates selected using station data, a percentage hit-rate is calculated, namely the amount of extreme events in station data divided by the amount of correctly selected dates in reanalyses.

## 4. Results

### 4.1 Spatial features and magnitude

While quantitative estimates of how the reanalysis products differ from station data will be shown later, we first show multi-decadal climatology and tendency maps for a more qualitative inspection of the snow representation in reanalyses. Starting with the recent period, Figure 1 shows the snow depth climatology over 1981-2010 for April,



October and November. Unsurprisingly, April displays the overall highest values.
Highest snow depths over Eurasia are located in northern Siberia along the 90° E
meridian. Elevated snow depths are also found over the Russian Far East and over
Kamchatka in particular. Both of the features displayed in the station data are also
represented by all reanalysis products. Overall, there is a broad agreement in the
position of high snow depth areas as well as the snow region boundaries. However,
ERA20C shows notably lower snow depths in northern Siberia, compared to ERA-
INTERIM-land and 20CRv2c, but the latter shows generally higher snow depth than
station data, especially in April and November.

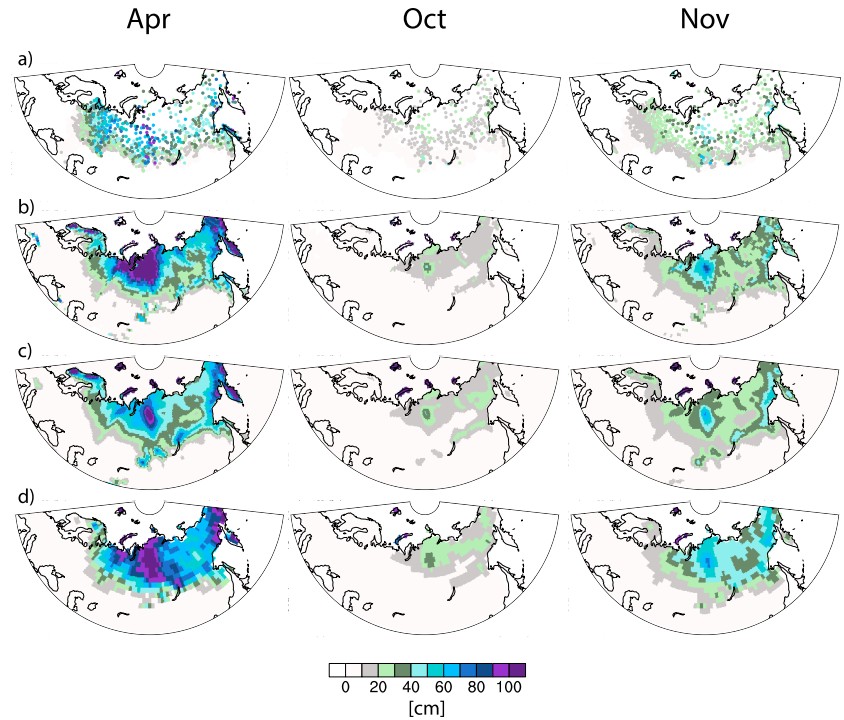


Figure 1: 1981-2010 snow depth climatology of (from left to right) April, October and
November in a) observations, b) ERA-INTERIM land-d c) ERA20C and d) 20CRv2c.
ERA20CL, ERA-INTERIM land-e and 20CRv2 are not displayed due to insubstantial
differences to ERA20C, ERA-INTERIM land-d and 20CRv2c.
The decadal tendency in the recent era is shown in Figure 2, as snow depth anomalies
between the 1996-2010 period minus those in the 1981-1995 period. In April, the



region with strongest snow depth decrease is the western, European part of Russia,
west of the Urals and between the Barents and Caspian Sea. This feature is clearly
underestimated by all reanalyses, best represented by 20CRv2, followed by ERAINT-
l. However, the sign of the tendency is not homogenous over the region in the
reanalyses, and local snow depth increases can be found. A second region of snow
decrease, which is broadly captured by the reanalyses is the Russian Far East, with
ERA20C displaying poorer agreement. A pronounced positive anomaly is found in
reanalyses north of Lake Balkhash and extending toward the coasts of the Bara and
Laptev Seas, a region where the station coverage is poor though. Towards southern
Russia, the observed signal is more complex with snow depth increase towards the
border to Kazakhstan, but with snow depth decrease further east on the western side
of Lake Baikal, which the gridded products fail to capture, both in terms of extend
and magnitude. In autumn, and especially in November, the in-situ data reveal a broad
longitudinal dipolar pattern with decrease (increase) of snow depths in the eastern
(western) part of Russia, reproduced by the reanalyses.
Overall, 20CRv2c captures the observed patterns slightly better than ERA-Interim-
land, while ERA20C shows the poorest agreement.



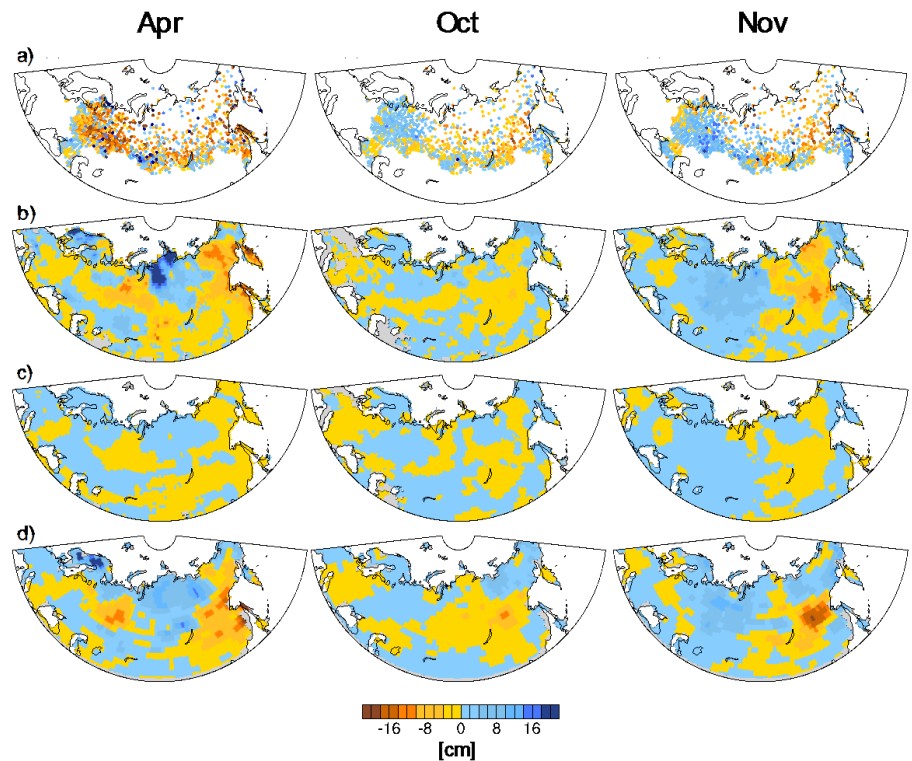

Figure 2: 1996-2010 minus 1981-1995 snow depth anomalies of (from left to right) April, October and November in a) observations, b) ERA-INTERIM land-**-d**, c) ERA20C and d) 20CRv2c. ERA20CL, ERA-INTERIM land-e and 20CRv2 are not displayed due to insubstantial differences to ERA20C, ERA-INTERIM land-d and 20CRv2c.

**4.2 Inter-decadal performance**

Figure 3 shows the long-term decadal changes over the Northern Russia snowpack (averaging between 50°-150° E and 60°-75° N) in the different climate reanalyses. Series of 30-year climatological anomalies were computed with a moving window of 10 years, using 1981-2010 period as a reference climatology. From the 1941-1970 period onward, all four products show similar tendencies. Further back in time however, the gridded products diverge: ERA20C & ERA20CL continue a downward tendency (mean anomalies decrease) whereas the 20CRv2 & 20CRv2c reanalyses show an overall increase in snow depth, resulting in a notable difference by the early




20th century. This evolution is, despite minor differences, true for all three months.
For all months, the 20CR family of reanalyses show strong positive anomalies for the
1911-1940 period, the main period of the Early Twenty Century Arctic Warming
(ETCAW).

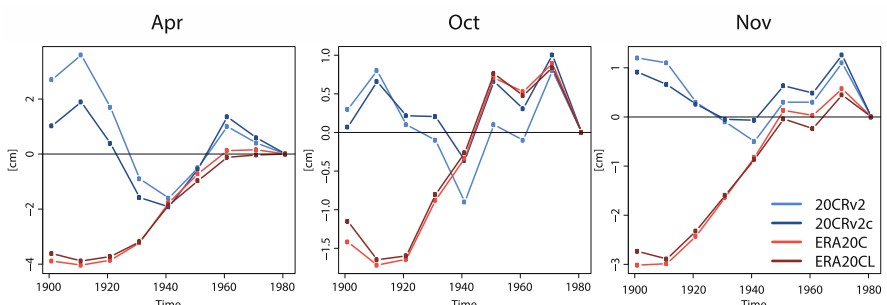


Figure 3: Time series of snow depth anomalies in (from left to right) April, October
and November averaged over the main northern Russia snow pack (50°-150° E, 60-
75° N). Each data point represents a 30-year long climatology, starting from 1901-
1930 until 1981-2010 with 10 year shifts. Anomalies are calculated relative to the
1981-2010 climatology.
Unfortunately, none of the 13 selected stations with a long record is located in that
northern Russia region. A similar behavior emerges however if the comparison is
made between the 13 stations and the collocated reanalysis data, as shown on Figure 4.
Again, comparing to the 1981-2010 reference climatology disregards differences in
snow depth magnitude and helps focusing on long-term tendencies. All three months
show a divergence of the two reanalysis families towards the beginning of the 20th
century. Going backward in time from the recent era, tendencies are similar until the
1941-1970 period but, afterwards, the ECMWF reanalyses show a declining mean
snow depth whereas the 20CR reanalyses favor an increase in snow depth.
Interestingly, snow station data agrees very well with the 20CR reanalyses until ca.
1951-1980 period, while the ECMWF reanalyses show much more pronounced
deviations from the station data anomalies. Towards the beginning of the century, the
station data agrees more and more with the ECWMF reanalyses in late autumn, but
20CRv2 is closer to station data in April. The ECMWF reanalyses achieve an
excellent representation for the 1901-1930 and 1911-1940 periods in autumn (for the
1901-1930 spatial anomalies see Supplementary Figure 2).



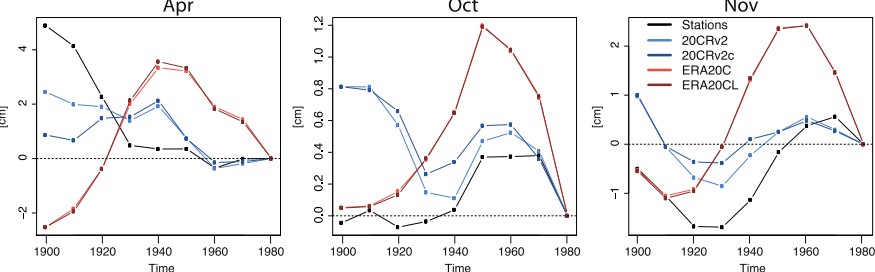


Figure 4: Top: Time series of snow depth anomalies in (from left to right) April, October and November for the average of the 13 station locations. Each data point represents a 30-year long climatology, starting from 1901-1930 till 1981-2010 with 10 year shifts. Anomalies are calculated relative to the 1981-2010 climatology.

291

## 4.3 Sub-decadal and daily performance

Moving away from decadal tendencies, we now evaluate the daily and the inter-annual snow variability over the 13 selected stations with records extending back to the early days of the 20th century. Figure 5 presents the daily performance between station data and the reanalyses over the recent period (1981-2010).

The melting season (April) generally exhibits the weakest correlation between grid and station, with slightly better values for October and highest values for November. However, this ranking can differ for individual station locations. For the period 1981-2010, the ERA20C reanalysis achieves better results than the 20CR reanalyses, especially so in April, indicating that melting and temperature evolution is somewhat more accurate in the ECMWF reanalyses. November and even more so October correlations are very similar in all four long-term reanalysis products. As to be expected, the ERA-INTERIM-land reanalysis, given the higher quality of atmospheric forcing in the recent era and the finer spatial resolution, generally scores the highest when compared to the respective station with medians above 0.8 in all three months. Note that in the correlation analysis ERA-INTERIM-land-d achieves higher averaged correlation coefficients than the uncorrected version.

Looking at long-term correlations (Figure 6), the ECMWF reanalyses slightly

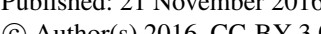



outperform the 20CR in April, but less so than in the 1981-2010 period. The opposite
is now true for October, where the 20CRv2 and 20CRv2c achieve slightly higher
averaged correlation coefficient values, whereas in November, all long-term
reanalyses have comparable correlations with station data with slightly higher values
for the 20CR family. In two out of three months, the ERA20C-land version does not
realize higher accuracy than the parent product ERA20C. The same is true for the new
20CRv2c, which outperforms 20CRv2 only in November.
We note that long-term daily correlation coefficients for individual northern stations
repeatedly exceed 0.7 (see Supplement Table 1). Only two stations (ID 30758 & ID
35121) consistently show very low correlations across the seasons and reanalyses,
probably because of their southern positions. In general terms, the linear correlation
performance decreases from northern to more southern stations. This reflects the
sensitivity of snowfall in relatively mild environments, resulting in short periods of
snow availability. Such small-scale snowfall events are hardly captured by the
reanalyses.

325  .

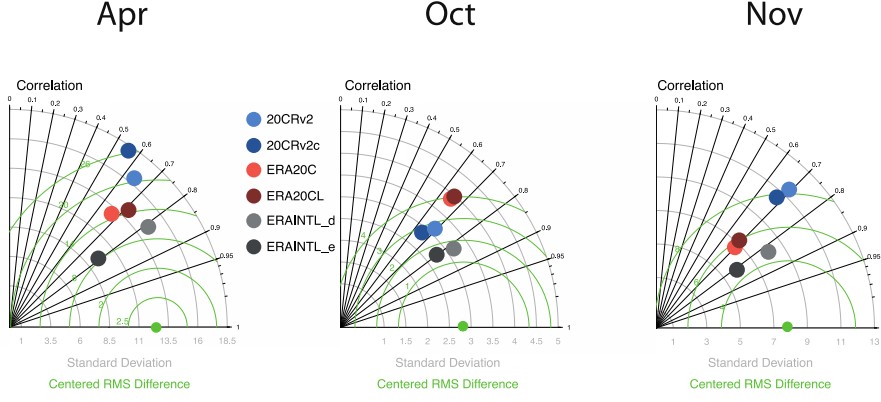

Figure 5: Taylor diagrams showing the median of the 13 station locations using daily
data for the period 1981-2010. The X-axis and Y-Axis indicate the standard deviation,
the radians indicate correlation values and the green circles indicate centered RMSE.
The green dot shows the observed variability. For more details concerning the
datasets statistics, see Supplementary Figures 3-5.




332 Root mean square error (RMSE) values obviously differ from location to location (see

333 supplement Table 1). Averaging over all stations reanalyses products were found to

334 produce the absolute largest deviations from the *true* station timeseries in April,

335 followed by November and lastly October. The low October RMSE is influenced by

336 the relatively small absolute snow depth values during that month. Thus, even

337 deviations from zero (e.g. incorrect event of snowfall) will be small. Again, as

338 expected the ERA-INTERIM land produces the smallest RMSE over all reanalyses.

339 The ERA-INTERIM land version without the precipitation correction has lower

340 RMSE in April and November than the version with the precipitation correction. This

341 could be due to the scarcity and uncertainty of rain-gauge observations in the region,

342 which would deteriorate the GPCP-based correction. The pair of ERA20C reanalyses

343 clearly outperforms the 20CR pair in April and November, but is on equal terms in

344 October.

345

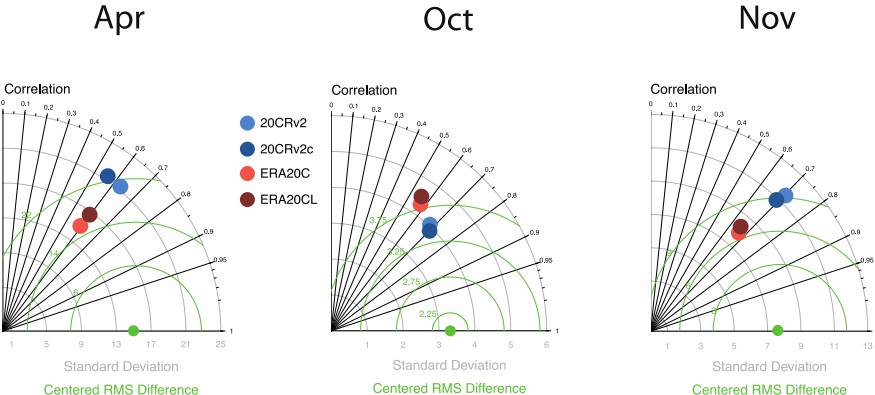

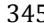

347 Figure 6: Taylor diagrams showing the median of the 13 station locations using daily

348 data for the longest period available (see Table 1). The X-axis and Y-Axis indicate the

349 standard deviation, the radians indicate correlation values and the green circles

350 indicate centered RMSE. The green dot shows the observed variability. For more

351 details concerning the datasets statistics, see Supplementary Figures 3-5.


353 Finally, to address variability characteristics of the reanalysed snow depth values,

354 Figure 5&6 (X-axis) also show the median standard deviation of anomaly time series



averaged over the 13 stations. As expected, April and November show much higher
variability than October. All ECMWF products show a good representation of the
station standard deviation. The uncorrected ERA-INTERIM land version apparently
suppresses a certain amount of variability with lower median values than the rest of
the ECMWF family products. On the other side, both 20CR reanalyses overestimate
the variability. October values for 20CRv2 and 20CRv2c are very much influenced by
one outlier location, so that the median is still well within the range of the station
median.
Assessment of variability is especially important in the framework of extreme events.
Since the replication of variability and daily correlation seems promising, an extreme
event hit-rate is computed to measure how well the reanalysis products can detect the
exact dates of extreme events. Figure 7a shows the hit-rate of days with extreme
absolute snow depth values whereas Figure 7b shows the hit-rate of days with
extreme accumulation of snow depth for the 13 station locations. Better daily
correlations in April (Fig. 5) seem to help the ERA20C reanalyses to capture slightly
more dates correctly than the two 20CR products. The opposite is true for autumn
months, especially for absolute snow depth maxima. Interestingly, changing from
absolute to accumulation extremes helps ERA20C to achieve a higher hit-rate,
whereas the 20CR products show a slightly worse hit-rate for the latter metric.
Moreover, ERA20C land, which shows a very similar if not better performance for
absolute snow depth extremes, shows a poorer performance for detecting
accumulation extremes. Overall though, mean hit-rates stay well below 40%; only for
single locations did the hit-rates exceed 50%.



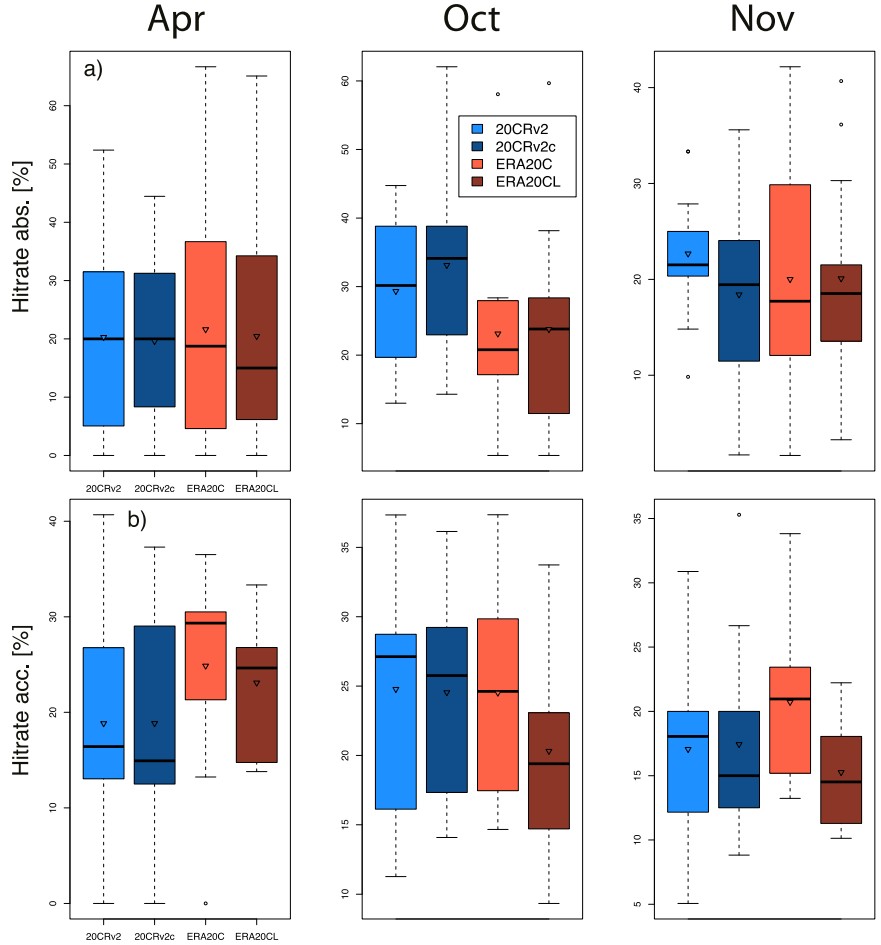

Figure 7: Boxplots graphs for the extreme events hitrate analysis of the 13 snow depth
station locations, where the triangle denotes the mean, the bold black line denotes the
median, the box denotes the 25-75% percentile range (or interquartile range), the
whiskers show the upper and lower end or at most the 1.5 x interquartile range and the
dots denote outlier. a) shows boxplots for absolute snow extreme events the longest
possible time period, b) same as a) but for snow accumulation.

**5. Discussion**

Comparing snow depths in multiple long-term, centennial reanalyses with in-situ
measurements over Russia, our results indicate a good performance of the reanalysis
datasets. Climatologies are well represented and long-term daily correlations revealed





very high coefficient values for most of the station locations. Snow depths from
surface input-only reanalyses consistently show linear correlations of 0.6 and higher,
although dealing with a very large sample size. Khan et al. 2008 found best case
basin-wide correlations of around 0.65 in ERA-40 and JRA-25, with much worse
correlations for the NCEP-DOE reanalysis. All these reanalyses assimilated a variety
of input data, not only surface data as is the case with the centennial reanalyses
examined in this study. Moreover, Khan et al. 2008 state that all evaluated reanalysis
snow products showed the worst matching in April.
The same result was found in our analysis, where April values showed the smallest
correlation and highest absolute error (RMSE). Therefore, it can be assumed that
models used for creating the reanalysis datasets still struggle with properly
representing melting season (Slater et al. 2001). Looking at the RMSE, it could be
shown that the 20CRv2 & 20CRv2c generally overestimate snow depth, and that
ERA20C & ERA20CL are closer to the station data. The same applies to the
variability comparison. Interestingly, the snow depth RMSE in October is smaller
than in the other months, but day-to-day variability (correlation) appears to be better
in November. This indicates that the initial snowfall in October, if occurring, is harder
to capture than in November, but also generates only small snow depths. Therefore,
even if completely missed by the reanalysis, it produced only small RMSEs.
Peings et al. 2013 found that 20CRv2 displays a good performance in detecting the
daily advance of October and November snow (between 80-100% hitrate). We found
that 20CRv2 shows good long-term daily correlations in October and November, even
higher than ERA20C. That said, binary snow information as well as correlation
analysis masks the details of snow amount, which is better seen in anomaly or
climatology maps. Moreover, our hit-rate analysis of dates for extreme snow depths
and snow accumulation showed that for the 13 station locations only about 40% of the
dates were correctly computed when compared to station data. Among the
explanations for this underwhelming performance are a) the assimilation of only
surface data in the reanalyses (which challenges the computation of the complex
conditions for extreme snowfall), b) the long time frame in which assimilated data
quantity is decreasing back in time and c) spatial resolution of the reanalyses which
can not resolve features like small scale uplift or orographic precipitation, or at even



smaller scale, snowdrift. With these deficiencies in mind, the achieved correlation
coefficients for the centennial timeseries are even more remarkable.
However, analysis of inter-decadal tendencies of snow depth revealed a peculiar
evolution. Generally, the ECMWF datasets compute a stronger snow depth decrease
before the 1940s than the 20CR products for the main Russian Arctic snow field.
Since climatological maps do not show substantial differences, origin of the large
disagreements must emerge in the pre-1950s period. The assimilated input data is near
identical between ERA20C and 20CRv2c, and thus model biases seem to be the
source of divergence.
One reason for the snow depth evolution could be the overestimation of Arctic SLP
(sea level pressure) during the pre-1950s in ERA20C (Belleflamme et al. 2015).
Indeed we found that ERA20C shows high (higher than 20CR or reconstructed
values) positive SLP anomalies for the beginning of the 20th century over Central
Russia (see Supplementary Figure 6). Such a high anomaly over the high latitudes
might lead to reduced poleward moisture transport, as well as decreased cloud cover
and downward long wave radiation, which is very efficient in melting snow.
Moreover, stable atmospheric conditions prevent vertical motion and therefore
condensation. Knudsen et al. 2015 showed that, in the recent era, both a positive SLP
anomaly and a negative anomaly in snowfall prevail over the Russian Arctic coast in
summer months with high sea ice melt. Hence, Arctic anti-cyclonic circulation
patterns that are associated with sea ice melt also promote low snowfall over the
Russian sector of the Arctic, and a similar association could be at play in ERA20C in
the pre-1950s. On the other hand, if compared to station data, the ERA20C snow
depths show a good agreement for anomalies early in the 20th century.
Furthermore, near-surface temperatures influence snow depth evolution. The new
20CRv2c dataset uses alternative sea ice and SSTs representations as boundary
conditions, which improves the 2m temperature performance over the Arctic
compared to 20CRv2. Nevertheless, it is generally still colder than ERA20C or
CRUTEMP. However, ERA20C is most probably much too warm during April,
whereas the 20CR reanalyses seem to be too cold during November and December,
thus they might be overestimating snow depths (see Supplementary Figures 7 and 8).
Ultimately, there is no clear and simple answer to this issue and our analysis can only





provide an initial assessment of the discrepancy between the two families of
reanalyses.
The results of the snow climatologies hint towards heterogeneous dataset issues.
Decadal tendencies in the second half of the 20th century are better represented by the
20CR datasets (relative to their baseline), whereas tendencies for the first half of the
century are better represented in ERA20C. Unfortunately, only 13 stations could be
used to verify long-term evolution in snow depth. Data recovery from a higher density
network with better spatial coverage is needed to really constrain the diverging snow
states in these long-term reanalyses. Moreover, future reanalysis or model
comparisons might be needed. The planned CERA (ERA20C plus coupled ocean) and
GSWP3 could give further insight into this topic. Model inter-comparisons
concerning snow representation might reveal necessary qualities to compute a realistic
snow depth.
**6. Conclusion**
Snow depth and its evolution from a variety of centennial reanalyses have been tested
against in-situ observations over the Russian territory. Long-term reanalyses are able
to reproduce daily and sub-decadal snow depth variability very well. That said,
computing the exact day of extreme snow accumulation is still a difficult task for
these datasets. Spatially, the region of high and low snow, and the snow cover
boundaries are well represented. However, inter-decadal comparison of snow depth
revealed some issues with pre-1950s snow climates over northern Russia. The
ECMWF and NOAA reanalyses show diverging snow states (low or high,
respectively), most probably likely a consequence of assimilation schemes or model
biases rather than input data.
To further understand and quantify changes during the current and future Arctic warm
periods, it is imperative to maintain and expand a dense network of (Arctic) snow
measuring stations (including their meta data). Reproducing observed snow (depth) in
climate models is a difficult challenge since many environmental factors determine
snowfall amount and ultimately snow depth. In-situ snow depth measurements and
reanalyses are important tools to evaluate the performance of climate models .





**Acknowledgments.** YO was supported by the Norwegian Research Council (project
SNOWGLACE # 244166 and EPOCASA #229774/E10). AS and SB were supported
by the EU-FP7 project ERA-CLIM2 (607029). MW, YO, SB, AS and OB
acknowledge funding by the European ERAnet.RUS programme, especially within
the project ACPCA.

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
