# Peer review of "Eurasian snow depth in long-term climate reanalyses"

_The Cryosphere, 2016_

## Referee Comment (RC1) · Anonymous Referee #1 · 9 Dec 2016

General comments: Overall, this a informative and relevant paper with some issues, which need to be resolved. The paper investigates the performance of different reanalyzes products in representing snow depth in the NE part of Eurasia. The authors use daily snow depth measurements from 820 Russian meteorological stations to compare climatologies and 13 long-term stations to analyze temporal differences. The topic of the investigation fits very well into the journal's scope. It is one of the very few studies that thoroughly evaluates the snow depth represented in different reanalyzes products. As such, I consider the work as being relevant for the scientific community. For most parts, the methods are described appropriately and the conclusions are well-based on the results obtained. The paper however suffers from a simple overview (look up table) of the underlying datasets. Please see the listing below for further details. These issues should be improved before publication of the paper. For this purpose, only very

few new analyses are required and the basic structure of the paper does not have to be changed. I'd therefore suggest returning the manuscript to the authors for minor revisions.

Major issues:

- For readers not familiar with reanalysis products a paragraph is missing where it is explained which snow variables are provided in such products and how they are calculated.

- A table is missing where the characteristics of the different reanalysis products are listed. Such a table should contain which product belongs to which of the two families, what are the differences in regard to the assimilated data, what are the differences in spatial and temporal resolution, etc.

- I miss a kind of uncertainty assessment. Could you please mention that there is some uncertainty due to the elevation differences between the grid cell and the station. Did you also try to use the neighboring grid cell with smallest elevation difference instead? The temporal resolution of the reanalysis products may also not fit the snow observation time. Do products with finer spatial or higher temporal resolution perform better?

- In order to be able to properly assess the different errors measures for the 15 long-term stations presented in different figures the reader needs to have an idea about the mean and standard deviation of the different analyzed snow depth values of each individual station. I suggest to add this information to table 1 or to add a new table. The information of the percentage of missing values is currently hard to read and could be easily combined with the climatological information of each station.

- In order to test if the relatively poor hitrate is influenced by temporal issues between reanalysis and observation, I suggest to also calculating the hitrate when +/- 1 day shift in the reanalysis is allowed.

Minor issues:

L30: On order to prevent misunderstanding, replace "data sets" with "reanalysis products"

L66: Why "slowly"? Often the state of the snow cover changes very fast.

L116-117: The last sentence in this paragraph cannot be understood by readers unfamiliar with reanalysis products.

L163-174: What is the difference to the "Historical Soviet Daily Snow Depth (HSDSD) product [Armstrong, 2001]"? Would there be more long-term data series than only 15?

L163-174: Please add some information how snow depth was measured. Point measurement on a stake or mean snow depth from snow courses? Just out of personal interest: What did change in the measurement procedure after 1965?

L192: I cannot find "red marked" stations?

L199: daily accumulated snow depth

L213-214: To be able to better follow your explanations, the meridians should be indicated in Figure 1.

L222-225: Please add a sentence mentioning that the depicted snow depth represents the mean maximum snow depth for each shown month.

L252-253: Please explain why you compare Northern Russia (and e.g. not Eastern Russia) in this step.

Figure 3: Is there any argument not to use the same scale on all three graphs?

Figure 4: Is there any argument not to use the same scale on all three graphs?

L293: "Daily" still means monthly maximum snow depth?

Figure 7: Is there any argument not to use the same scale on all six graphs? Are these

hitrates based on 1981-2010 or on the longest period available?

L388-390: Please add links to the table and figure where these results can be seen. What are the arguments to call the correlations "very" high? They are mostly below 0.8.

L391: I don't understand what you mean with "although dealing with a large sample size"?

L400-402: To which period does this statement apply?

L403-404: I guess the RMSE is smallest in October because absolute values are smallest in October!

L449: Crutemp: Please add version and reference.

Supplementary Table 1: What period do these numbers refer to?

Supplementary Figure 3-5: Is there any argument not to use the same scale on all three graphs?

Should the median values of Supplementary Table 1 not be found in Figure 3 and 4?

Supplementary Figure 5: The unit "cm2" for the variance seems strange? Why not use the more common measure standard deviation?

Supplementary Figure 6: Is there any argument not to use the same scale on all three graphs? What is Hadsipr2?

Supplementary Figure 7: Is there any argument not to use the same scale on all three graphs?

---

## Referee Comment (RC2) · Anonymous Referee #2 · 19 Jan 2017

The manuscript addresses an important topic, which fits very well to the scope of the journal. There has been a lot of uncertainty in the recent trends in Siberian snow cover in autumn, and the manuscript to some degree reduces this uncertainty, by showing that the observed trends strongly vary in space (Figure 2a). Moreover, interesting results are presented on the centennial time scale, showing major differences between the U.S. and European reanalyses until about 1940. The manuscript has, however, also weaknesses, and I suggest that major revisions should be made before publication.

Major comments:

1. A lot of results are presented on the performance of reanalyses in various months and regions, evaluated using various skill scores. The manuscript is, however, lacking analysis on the reasons for the better of worse performance of reanalyses. For example, major differences are found for the period 1901-1940 (Figures 3 and 4), and a reader is certainly interested in understanding the reasons for the differences. The differences can originate from (a) different data assimilated or different methods applied in assimilation of the same data, (b) different model results for precipitation and its phase, (c) different model results for snow melt, and possibly (d) different parameterizations (if any) applied for snow metamorphosis causing changes in snow density and, accordingly, thickness. The authors should pay at least some attention on these issues. If it is too difficult to find answers to issues (b) to (d), at least the snow schemes applied in the models should be compared. There may be major differences in the schemes for snow thermodynamics, which may explain the different results in early years when the role of data assimilation was probably smaller.

2. The arguments for conclusions presented in Sections 5 and 6 are not clear. Why do you write in the beginning of Section 5 that the results indicate a good performance of reanalyses (change "datasets" to "products") and that climatologies are well represented? All figures presenting comparisons against observations include considerable errors, and Figure 3 only comparing different reanalyses includes huge differences. Also, most of the correlation coefficients presented are not "very high". A correlation of 0.6 only explains 36% of the variance. If you consider the results good, did you have reasons (in addition to Khan et al. 2008) to expect worse results? Do you have arguments to set relevant thresholds for "good performance"?

3. In general, the text is not particularly clearly written. See Minor comments below.

Minor comments:

Lines 31-34: unclear text

Line 51: alter . . . modulate

Line 59: has severely impacted

Line 60: "From 1979 to 2011" or "Between 1979 and 2011"

Lines 62-63: I am not sure, if Park et al. (2013) also report regional snow cover increase associated with low sea ice concentration. The main message of their study is, however, the opposite, given by the title of the paper: "The role of declining Arctic sea ice in recent decreasing terrestrial Arctic snow depths".

Line 76: climate models

Lines 79-81: Global reanalyses have at least equally large spatial coverage as satellite products. So, the work "compromise" is perhaps not the best.

Lines 85-86: not all reanalyses listed here extend further back in time.

Line 98 and analogously in many other places: Brun et al. (2013)

Line 124: Medium-Range

Line 130: assimilating synoptic observations of atmospheric surface pressure

Line 144 delete "model"

Line 146: tell the resolution also in km.

Line 150: "follows exactly the CMIP5 proposal" is unclear

Line 186: perhaps "exceeding"

Lines 279-284: the text is unclear and appear contradicting. Be clearer to which seasons you refer to in the beginning. On lines 283-284 the ECMWF is considered excellent in 1901-1940, but in Figure 4 the ECMWF appear excellent only in 1901-1910 and 1980.

Lines 419-421: Snow drift may indeed generate differences between observations and reanalysis products. In addition to resolution, however, the differences may simply originate from a lack of snow drift parameterization in the reanalysis snow scheme (see Major comment 1).

Lines 427-428: The differences in input data should be quantified in Section 2.1.
Lines 438-443: The cause and consequence related to sea ice melt remains unclear. Without clarifying this, the processes at play in the pre-1950s sound very speculative.

Line 449: Why do you think that ERA20C is most probably much too warm in April?

---

## Author Comment (AC1) · 16 Feb 2017

General comments: Overall, this a informative and relevant paper with some issues, which need to be resolved. The paper investigates the performance of different reanalyzes products in representing snow depth in the NE part of Eurasia. The authors use daily snow depth measurements from 820 Russian meteorological stations to compare climatologies and 13 long-term stations to analyze temporal differences. The topic of the investigation fits very well into the journal's scope. It is one of the very few studies that thoroughly evaluates the snow depth represented in different reanalyzes products. As such, I consider the work as being relevant for the scientific community. For most parts, the methods are described appropriately and the conclusions are well-based on the results obtained. The paper however suffers from a simple overview (look up table) of the underlying datasets. Please see the listing below for further details. These

issues should be improved before publication of the paper. For this purpose, only very few new analyses are required and the basic structure of the paper does not have to be changed. I'd therefore suggest returning the manuscript to the authors for minor revisions.

Response: We thank the reviewer for the support and valuable comments and suggestions that we address in detail below. The overhaul suggestions improve the clarity of the manuscript.

Major issues: - For readers not familiar with reanalysis products a paragraph is missing where it is explained which snow variables are provided in such products and how they are calculated.

R : Indeed, this information was missing. We added a paragraph about snow computation in the reanalyses at the end of section 2.1

- A table is missing where the characteristics of the different reanalysis products are listed. Such a table should contain which product belongs to which of the two families, what are the differences in regard to the assimilated data, what are the differences in spatial and temporal resolution, etc.

R : Thanks for this suggestion. A good overview table was needed. We added Table 1 in the manuscript with details concerning the differences in the reanalyses.

- I miss a kind of uncertainty assessment. Could you please mention that there is some uncertainty due to the elevation differences between the grid cell and the station. Did you also try to use the neighboring grid cell with smallest elevation difference instead? The temporal resolution of the reanalysis products may also not fit the snow observation time. Do products with finer spatial or higher temporal resolution perform better?

R : Thank you for pointing that out. We mention this uncertainty now in line 209. We did not include gridboxes with the smallest elevation difference since especially in the

case of 20CR, topography is rather coarse in the model and we wanted to keep the procedure the same in all gridded datasets. We also added in line 228 the information about temporal resolution. Indeed, observation time does not fit 100 % the daily (or 6-hourly resolution) in the reanalyses. We used a finer grid in ERA20C than was used for 20CR and we only see minor improvements. Assimilated data and model physics play a more important role.

- In order to be able to properly assess the different errors measures for the 15 long-term stations presented in different figures the reader needs to have an idea about the mean and standard deviation of the different analyzed snow depth values of each individual station. I suggest to add this information to table 1 or to add a new table. The information of the percentage of missing values is currently hard to read and could be easily combined with the climatological information of each station.

R : For better assessment of missing data we averaged the missing data for all three months and changed Table 2. Standard deviation can be seen in the Taylor diagrams. For mean value investigation, we initially had Figure 1 in mind. However, we see the point Reviewer #1 makes and added additional standard deviation and mean value analysis boxplots for the 13 long-term stations in the supplement, so it is easier for the reader to access these values.

- In order to test if the relatively poor hitrate is influenced by temporal issues between reanalysis and observation, I suggest to also calculating the hitrate when +/- 1 day shift in the reanalysis is allowed.

R : Very good point. We exchange this analysis in the manuscript and mention the results for the fixed date just briefly.

Minor issues: L30: On order to prevent misunderstanding, replace "data sets" with "reanalysis prod- ucts"

R : Changed

L66: Why "slowly"? Often the state of the snow cover changes very fast.

R : Changed to "corresponding"

L116-117: The last sentence in this paragraph cannot be understood by readers unfamiliar with reanalysis products.

R : Clarified L116

L163-174: What is the difference to the "Historical Soviet Daily Snow Depth (HSDSD) product [Armstrong, 2001]"? Would there be more long-term data series than only 15?

R : The dataset we used contains overall more stations, but the long-term stations are mostly the same. Therefore, unfortunately no more long-term data series than just 15.

L163-174: Please add some information how snow depth was measured. Point measurement on a stake or mean snow depth from snow courses? Just out of personal interest: What did change in the measurement procedure after 1965?

R : We added that information in section 2.2. The procedure of snow observations changed in the past: size of the stake (1924,1939) rules for the use of stake (1935, 1939), requirements for observation platform (1940, 1954), in the rules archiving (1966)

L192: I cannot find "red marked" stations?

R : That was an artifact, deleted

L199: daily accumulated snow depth

R : Changed

L213-214: To be able to better follow your explanations, the meridians should be indicated in Figure 1.

R : For clearer assessment of Figures 1&2, we added meridians.

L222-225: Please add a sentence mentioning that the depicted snow depth represents

the mean maximum snow depth for each shown month.

R : Added that information

L252-253: Please explain why you compare Northern Russia (and e.g. not Eastern Russia) in this step.

R : We use this area based on the climatology maps of snow cover. In our view this is the region with the highest snow depths. We added that explanation to the text.

Figure 3: Is there any argument not to use the same scale on all three graphs?

R : No there is not. We now use the same scale in all three graphs.

Figure 4: Is there any argument not to use the same scale on all three graphs?

R : No there is not. We now use the same scale in all three graphs.

L293: "Daily" still means monthly maximum snow depth?

R : Daily in this case means Âń as measured Âż, on a day to day basis. These are used for all following analysis procedures, like correlation, hitrate etc. This allows us to have a very strong statistic.

Figure 7: Is there any argument not to use the same scale on all six graphs? Are these hitrates based on 1981-2010 or on the longest period available?

R : No there is not. We now use the same scale in all three graphs. Hitrates are based in the longest period possible. We added that statement to the text.

L388-390: Please add links to the table and figure where these results can be seen. What are the arguments to call the correlations "very" high? They are mostly below 0.8.

R : Thanks for pointing that out. If we have the daily resolution and spatial grids in mind, the results are quite remarkable. However, our wording here was wrong. We changed the wording accordingly.

L391: I don't understand what you mean with "although dealing with a large sample size"?

R : Again, thanks for pointing that out. Wording was not correct. Is changed.

L400-402: To which period does this statement apply?

R : Longest period possible

L403-404: I guess the RMSE is smallest in October because absolute values are smallest in October!

R : We agree! This point is made in L406.

L449: Crutemp: Please add version and reference.

R : Reference and version is added.

Supplementary Table 1: What period do these numbers refer to?

R : Longest period possible, except for ERA-Interim where they refer to 1981-2010

Supplementary Figure 3-5: Is there any argument not to use the same scale on all three graphs?

R : For the boxplot graphs in the supplement we decided to keep different scales since metrics change the scale quite a bit between different months and we want reader to see the maximum amount of details since the Taylor Plots show only median values.

Should the median values of Supplementary Table 1 not be found in Figure 3 and 4?

R : Thank you for pointing that out. Yes, they should be found in these figures. However, we found that the numbers for ERA-Interim were not up to date (wrong time window selected) and there was an error in one entry for ERA20c-land. We updated all numbers accordingly.

Supplementary Figure 5: The unit "cm2" for the variance seems strange? Why not use

the more common measure standard deviation?

R : We added a boxplot for standard deviation

Supplementary Figure 6: Is there any argument not to use the same scale on all three graphs? What is Hadsipr2?

R : No there is not. We now use the same scale in all three graphs. We added the information about the SLP reconstruction

Supplementary Figure 7: Is there any argument not to use the same scale on all three graphs?

R : No there is not. We now use the same scale in all three graphs.

---

## Author Comment (AC2) · 16 Feb 2017

The manuscript addresses an important topic, which fits very well to the scope of the journal. There has been a lot of uncertainty in the recent trends in Siberian snow cover in autumn, and the manuscript to some degree reduces this uncertainty, by showing that the observed trends strongly vary in space (Figure 2a). Moreover, interesting results are presented on the centennial time scale, showing major differences between the U.S. and European reanalyses until about 1940. The manuscript has, however, also weaknesses, and I suggest that major revisions should be made before publication.

Response: We thank the reviewer for his insight and for raising a few key questions regarding the way we describe and discuss the results. We present below a detailed reply.

Major comments: 1. A lot of results are presented on the performance of reanalyses in various months and regions, evaluated using various skill scores. The manuscript is, however, lacking analysis on the reasons for the better of worse performance of reanalyses. For example, major differences are found for the period 1901-1940 (Figures 3 and 4), and a reader is certainly interested in understanding the reasons for the differences. The differences can originate from (a) different data assimilated or different methods ap- plied in assimilation of the same data, (b) different model results for precipitation and its phase, (c) different model results for snow melt, and possibly (d) different parame- terizations (if any) applied for snow metamorphosis causing changes in snow density and, accordingly, thickness. The authors should pay at least some attention on these issues. If it is too difficult to find answers to issues (b) to (d), at least the snow schemes applied in the models should be compared. There may be major differences in the schemes for snow thermodynamics, which may explain the different results in early years when the role of data assimilation was probably smaller.

R : Thank you for pointing out some key elements. In the discussion part we investigate several options why the difference might occur, namely temperature and sea level pressure differences between the datasets. However, as you rightly pointed out, an outline of differences among the snow scheme was missing, which is added now at the end of section 2.1 . Moreover, we added Table 1, where it is more apparent what data assimilation is used and what boundary conditions are used. That said, we can dig only so far into technical details. Our investigations still show that assimilation and snow schemes are very similar, and we still support the idea of dynamical reasons for the changes in snow. We added a plot for vertical integrated mass of atmosphere, which points out a problem in ERA20C, namely to much high pressure over the Arctic in the first half of the 20th century. With this we hope to give enough initial ideas as to why the snow states diverge. Future studies need to check this feature in more detail.

2. The arguments for conclusions presented in Sections 5 and 6 are not clear. Why do you write in the beginning of Section 5 that the results indicate a good performance

of reanalyses (change "datasets" to "products") and that climatologies are well represented? All figures presenting comparisons against observations include considerable errors, and Figure 3 only comparing different reanalyses includes huge differences. Also, most of the correlation coefficients presented are not "very high". A correlation of 0.6 only explains 36% of the variance. If you consider the results good, did you have reasons (in addition to Khan et al. 2008) to expect worse results? Do you have arguments to set relevant thresholds for "good performance"?

R : Indeed, the wording here is not correct. We clarified the section and added arguments as why we see the performance as "good"

3. In general, the text is not particularly clearly written. See Minor comments below. Minor comments: Lines 31-34: unclear text

R : clarified

Line 51: alter . . . modulate

R : changed

Line 59: has severely impacted

R : changed

Line 60: "From 1979 to 2011" or "Between 1979 and 2011"

R : changed

Lines 62-63: I am not sure, if Park et al. (2013) also report regional snow cover increase associated with low sea ice concentration. The main message of their study is, however, the opposite, given by the title of the paper: "The role of declining Arctic sea ice in recent decreasing terrestrial Arctic snow depths".

R : Indeed, they only report regional specifics. Deleted the citation at this point

Line 76: climate models

R : changed

Lines 79-81: Global reanalyses have at least equally large spatial coverage as satellite products. So, the work "compromise" is perhaps not the best.

R : Clarified

Lines 85-86: not all reanalyses listed here extend further back in time.

R : Clarified

Line 98 and analogously in many other places: Brun et al. (2013)

R : corrected

Line 124: Medium-Range

R : corrected

Line 130: assimilating synoptic observations of atmospheric surface pressure

R : corrected

Line 144 delete "model"

R : deleted

Line 146: tell the resolution also in km.

R : We added resolution information in Table 1

Line 150: "follows exactly the CMIP5 proposal" is unclear

R : Not sure what is unclear at this point. Added explanation as to what is CMIP5.

Line 186: perhaps "exceeding"

R : changed

Lines 279-284: the text is unclear and appear contradicting. Be clearer to which seasons you refer to in the beginning. On lines 283-284 the ECMWF is considered excellent in 1901-1940, but in Figure 4 the ECMWF appear excellent only in 1901-1910 and 1980.

R : We tried to clarify that part. However note that each point represents a 30 year long climatology, which is shifted by 10 years from point to point rather than a 10 year long climatology.

Lines 419-421: Snow drift may indeed generate differences between observations and reanalysis products. In addition to resolution, however, the differences may simply originate from a lack of snow drift parameterization in the reanalysis snow scheme (see Major comment 1).

R : We added information about snow schemes in Section 2.1, see above.

Lines 427-428: The differences in input data should be quantified in Section 2.1.

R : Input data is now part of Table 1

Lines 438-443: The cause and consequence related to sea ice melt remains unclear. Without clarifying this, the processes at play in the pre-1950s sound very speculative.

R : We clarified this part. We do not want to tackle sea ice feedbacks here. This example should just be used as a dynamical reason as to why high pressure can lead to less snowfall.

Line 449: Why do you think that ERA20C is most probably much too warm in April?

R : Our best guess are dynamical reason, like temperature advection, due to pressure differences.